# A New Genus and Species of Gall-Forming Fordini (Hemiptera: Aphididae) on *Rhus wilsonii* Hemsl. from Yunnan, China

**DOI:** 10.3390/insects13121104

**Published:** 2022-11-30

**Authors:** Catherine Hébert, Xin Xu, Zixiang Yang, Colin Favret

**Affiliations:** 1Department of Biological Sciences, University of Montreal, Montreal, QC H1X 2B2, Canada; 2Institute of Highland Forest Science, Chinese Academy of Forestry, Kunming 650224, China

**Keywords:** Anacardiaceae, aphid, Eriosomatinae, gall, *Qiao* gen. nov., *Qiao jinshaensis* sp. nov., Sternorrhyncha, sumac

## Abstract

**Simple Summary:**

A new species of aphid, causing the formation of large galls on a sumac species endemic to Yunnan, China, is named and described. Molecular and morphological diagnostic criteria are provided, and digital specimen data made available.

**Abstract:**

A new species of gall-forming aphid from China, *Qiao jinshaensis* gen. et sp. nov., is described from *Rhus wilsonii* Hemsl. Morphological identification and molecular analyses both support the establishment of a new genus. A diagnosis combining morphological and molecular characters from alate viviparae is provided and specimen metadata are published in an open-access and machine-readable format.

## 1. Introduction

In China, *Rhus*-feeding aphids (Hemiptera: Aphididae: Eriosomatinae: Fordini) are of economic importance for the tannins extracted from the galls they produce [1]. There are currently 13 aphid species, in six genera, known to colonize only six species of *Rhus* (Anacardiaceae) [2,3]. Different species can make distinctly shaped galls on the same plant host. Phylogenetic analyses support the monophyly of most genera, with the exceptional placement of *Nurudea ibofushi* Matsumura, 1917 and *Meitanaphis flavogallis* Tang, 1978, which are repeatedly placed within *Schlechtendalia* Lichtenstein, 1883 and *Kaburagia* Takagi, 1937 respectively [4,5,6,7].

Aphid galls were collected for the first time on *Rhus wilsonii* Hemsl. in Yunnan, China. Their shapes were similar to those formed by some *Nurudea* Matsumura, 1917 and *Floraphis* Tsai & Tang, 1946 species, but the morphological identification of the aphids did not correspond to any currently known species. Morphological and molecular analyses were conducted but were not conclusive in assigning the species to any existing genus, thus supporting the establishment of a new one. We here formally establish *Qiao jinshaensis* gen. et sp. nov. Morphological and molecular diagnostic criteria are used following the protocol in Hébert et al. [8]. Additionally, the specimen collection data are published in a machine-readable format, making them readily accessible for future research.

## 2. Materials and Methods

### 2.1. Field Sampling

Fresh galls were collected in Yunnan, China (28.014° N, 103.638° E) on the host plant, *Rhus wilsonii*, and taken to the lab (see below for full collection data). Aphids were extracted from the galls and immersed in 70% ethanol for slide preparation and 100% ethanol for DNA extraction.

### 2.2. Molecular Analysis

Total genomic DNA was extracted from 20–30 individual aphids collected from a gall using Ezup animal genome DNA extraction kit at Sangon Biotech Co., Ltd., (Shanghai, China), according to the manufacturer’s protocol. The standard primer pair EF2 and EF3 [9] were used to amplify the elongation factor 1-alpha (EF1α) gene and HCO2198 and LCO1490 [10] were used to amplify a region of the mitochondrial cytochrome c oxidase subunit 1 (COI) gene. Polymerase chain reaction (PCR) amplification was carried out using standard protocols [11] with annealing temperatures of 56 °C for the EF1α locus and 55 °C for the COI locus. To ensure the accuracy of the sequence, the EF1α target fragment was incised and purified using a gel extraction kit (EZ-10 Column DNA Purification Kit, Bio Basic Inc., Markham, ON, Canada) according to the manufacturer instructions. Purified PCR products were sequenced in both directions on an ABI-3730XL gene sequencer at Sangon Biotech Co., Ltd. (Shanghai, China). The two sequences have been deposited in GenBank (accession numbers OP536221 and OP548621).

Introns in the EF1α sequence were removed in Geneious Prime 2022 software (Biomatters Ltd., Auckland, New Zealand) for analysis. Simple genetic p-distances were calculated with a Clustal Omega multiple alignment [12], as implemented in Geneious, with EF1α and COI sequences from other *Rhus*-gall aphids (see Table 1 for accession numbers). Simple (single nucleotide) and compound (multiple nucleotides) molecular diagnostic characters were retrieved manually for the COI sequence. Nucleotide positions are in reference to the COI alignment with the *Acyrthosiphon pisum* (Harris, 1776) mitochondrial genome (accession FJ411411).

The tannin content of the new galls was measured by ultraviolet spectrometry [13,14].

### 2.3. Morphological Examination

Fourteen aphids were cleared and slide-mounted in Canada balsam [15]. The identification key of Blackman and Eastop [2], as well as the original published descriptions of other *Rhus*-gall aphids [16,17,18,19], were used to assess the identity of the new species. Measurements and pictures were taken under brightfield conditions with a Zeiss M2 AxioImager microscope, an AxioCam HRc camera, and Zen 2012 Software, version 1.1.1.0 (Carl Zeiss AG, Oberkochen, Germany). All measurements are in micrometers (μm).

Morphological abbreviations are as follows: BL—body length (measured from the frontal margin of the head to the end of the cauda); BW—basal width; ANT I, II, III, IV, V, VI—antennal segments I, II, III, IV, V, VI or their length; PT—processus terminalis of the last antennal segment or its length; BASE—base of the last antennal segment or its length; URS—ultimate rostral segments (IV + V) or their length; LMF—length of the metafemur; LMT—length of the metatibia; WMT—width of the metatibia (measured in the middle); LMTS—length of the longest setae of the metatibia; HT I—first segment of the hind tarsus or its length (measured on the ventral side); HT II—second segment of the hind tarsus or its length.

## 3. Results

### 3.1. Molecular Analysis

The COI genetic p-distances between the new species and other species of *Rhus*-feeding Fordini ranged from 9.4% to 12.4%, much higher than the mean 7.3% infrageneric divergence found among aphids in general [20], and within the 9.7–11.3% range of intergeneric divergence we measured between *Rhus*-feeding Fordini (Table 1). Additionally, the average sequence divergence between the new species and other *Rhus*-feeding genera was 10.6%, greater than the 9.5% divergence between the other genera excluding the new species. The EF1α sequence divergence between the new species and other *Rhus*-feeding Fordini ranged from 5.1% to 7.7%. The average EF1α sequence divergence between the new species and *Rhus*-feeding genera was 6.1%, much higher than that between just the other genera (excluding the new species), which was 3.8%. These data confirm that the new species is genetically distinct, at the generic level, from all other *Rhus*-galling aphids for which DNA sequence data are available.

### 3.2. Taxonomic Treatment

The genus and species are established together, in accordance with Article 13.4 of the International Code of Zoological Nomenclature (4th Edition) [21].


***Qiao* gen. nov.**


urn:lsid:zoobank.org:act:F2600ADA-F5A0-4478-BECC-F108676F11AE


**Qiao jinshaensis sp. nov.**


urn:lsid:zoobank.org:act:DFF64645-6FD5-4F86-869C-9CFADD376D66

#### 3.2.1. Types

The type species of the genus *Qiao* gen nov. is *Q. jinshaensis* sp. nov. The holotype of the *Q. jinshaensis* is deposited at the Institute of Zoology of the Chinese Academy of Sciences (IOZ(E)) with the catalog number IOZ(E)54809 (Figure 1): alate viviparous female: China: Yunnan Province: Yongshan County: Xiluodu Town; 28.014° N, 103.638° E, 1005 m elev.; coll. 2021-09-27; YANG Zixiang and XU Xin *leg*.; in gall of *Rhus wilsonii* Hemsl. Paratypes are deposited at the IOZ(E) and the University of Montreal Ouellet-Robert Entomological Collection (QMOR): 13 alate viviparous females, same data as holotype, IOZ(E)54810-54811, QMOR66201-QMOR66204, QMOR66206-QMOR66212. These same specimen collection data are provided in machine-readable format by Hébert et al. [22].

#### 3.2.2. Diagnosis

The new genus can be distinguished from other *Rhus*-galling aphids by several characteristics. It has six antennal segments compared to five in *Nurudea* and *Schlechtendalia*. It differs from *Meitanaphis* Tsai & Tang, 1946, by having a short and oblique stigma on the forewing and three (rarely two) hamuli on the hindwing costa (stigma long and sickle-shaped and two (sometimes one) hamuli in *Meitanaphis*). The numerous ring-like sensoria of *Floraphis* and *Melaphis* Walsh, 1867, differentiate them from our new genus, which has large, oblong sensoria. Lastly, *Kaburagia* has only one large oval sensorium on ANT III-VI, whereas our species has more variation in shapes and numbers of sensoria, with the larger ones on ANT IV-VI enclosing small, sclerotized islands. Thirteen simple and four compound nucleotide characters were unique to our new species when the COI sequence was compared with the other available aphid species causing galls on *Rhus* (Table 2).

#### 3.2.3. Description

Color. In prepared specimens: Head, antennae, thorax and legs medium-brown (Figure 2). Genital plate, anal plate, and cauda light brown. Abdomen very light, almost transparent.

Head. Triommatidium distinct (Figure 3). Pale, linear ventral depression from the middle ocellus to the clypeus (Figure 4). Front margin of the head with two setae, which are 1.25–1.96 × the BW of ANT III. There are 5–6 pairs of dorsal setae and 11–13 ventral setae. There are two to four wax glands. Non-sclerotic region at the insertion of the antennae. Rostrum reaching past the front coxae, with URS 0.55–0.69 × HT II and bearing two accessory setae. Antennal tubercles undeveloped. Antennae are six-segmented (Figure 5). ANT 0.24–0.32 × BL. ANT III and V subequal in length; ANT VI longest. Short PT, 0.09–0.16 × BASE with four apical setae. Sensoria irregular in shape and numbers. ANT IV-V-VI each with one large sensorium occupying most of the length of the segment, enclosing small, scattered islands of cuticle and sometimes accompanied by a smaller round or oval secondary sensorium. Secondary sensoria (between one and four) of ANT III with more shape variation, either as small or large transverse bands, with or without sunken grooves, round or oval. They can also vary between the two antennae of the same specimen.

Thorax. Wax glands present. Mesosternal furca ‘Y’-shaped with a well-developed stem (Figure 6). Trochanter and femur fused (Figure 7). LMF 3.86–4.69 × ANT III. Legs slender with short and pointed setae. LMT 0.28-0.37 × BL and LMTS 0.47–0.60 × WMT. HT I triangular, 0.25–0.36 × HT II (Figure 8). First tarsal chaetotaxy: 3-4-4, with distal margin forming small spines. Forewing with two cubital veins well separated at the base, media simple with basal half obsolete, stigma short and oblique, veins fading at their apical extremity (Figure 2). Hindwing with 3 (rarely 2) hamuli on the costal margin and two oblique veins, the outer one shorter and strongly curved inward.

Abdomen. Marginal and spinal wax glands (Figure 9). Cauda semicircular, length 0.35–0.57 × its BW, bearing three to four setae. Genital plate bearing 24–33 setae, mostly distributed on the apical margin. Spiracles inconspicuous. Siphunculi absent. Measurements presented in Table 3.

#### 3.2.4. Etymology

The genus *Qiao* is named for QIAO Gexia, eminent Chinese aphidologist at the Institute of Zoology of the Chinese Academy of Sciences. The name is feminine in grammatical gender, thus the adjective *jinshaensis* is also feminine; the masculine and neuter forms are *jinshaensis* and *jinshaense*, respectively. The specific epithet is derived from the aphid’s type locality in Yongshan County, along the Jinsha River, tributary of the Yangtze River. The aphid’s host plant, *Rhus wilsonii*, is mainly distributed at a few narrow areas along the Jinsha River.

#### 3.2.5. Biology

The galls are yellow-green or reddish in color and inflorescence-like in shape (Figure 10). They possess multiple cavities and are positioned on the apical part of the stem. The gall measures approximately 110 by 90 mm and resembles the one formed by *Nurudea shiraii* (Matsumura, 1917). The autumn migrants of the aphid appear in late September. The tannin content of the gall was 52.3%, higher than the 35.2% of galls formed by *N. shiraii* [23].

#### 3.2.6. Identification Key to Aphid Species on *Rhus* spp.

The following couplets should be inserted into the key to aphid species feeding on *Rhus* spp. as currently presented at Aphids on the World’s Plants [2].

6ANT III-VI each with a single large oblong rhinarium occupying most of its surface area ……*Kaburagia rhusicola*-All or most of ANT III-VI with more than one single rhinarium ……**6A**6ARhinaria on ANT III-VI varying in size, shape and number (some round or in transverse bands, others large sheets enclosing small, scattered islands of cuticle) ……*Qiao jinshaensis* sp. nov.-ANT III-VI with only separate, transversely elongate rhinaria ……**7**

## 4. Conclusions

Based on gall morphology alone, *Qiao jinshaensis* gen. et sp. nov. is most similar to species of *Nurudea* (see Figure 3 in Ren et al., 2018 [24]). However, it is quite distinct genetically and morphologically, suggesting the genus diverged early in the diversification of the *Rhus*-galling aphids. Despite this apparent divergence, these aphids, sometimes collectively considered the subtribe Melaphidina [5], are not especially speciose (there are now 17 species in 7 genera, including some species without a known host [25]). The genus *Rhus* dates to approximately 50 mya and is closely related to *Pistacia* [26], which is also known to harbor several genetically distinct but small *Fordini* genera. Price (2005) suggested that there may have been an early diversification among the *Pistacia*-galling aphids, now evident at the generic level, but with reduced recent speciation [27]. This hypothesis is also applicable to the *Rhus*-galling aphids. So far, most studies have focused on *Rhus*- or *Pistacia*-feeding groups separately (e.g., [7,28]), but given the evolutionary proximity of both the aphids and their hosts, a phylogenetic evaluation of the entire tribe is needed.

## Figures and Tables

**Figure 1 insects-13-01104-f001:**
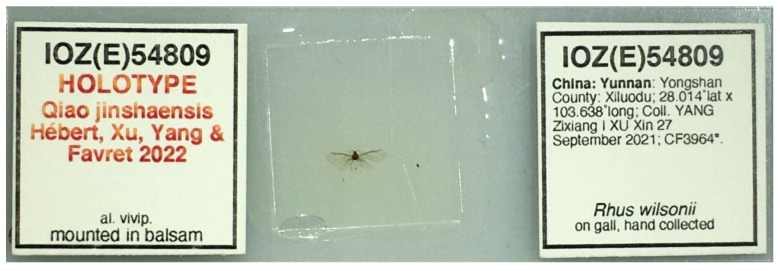
Holotype slide of *Qiao jinshaensis* sp. nov. (IOZ(E)54809).

**Figure 2 insects-13-01104-f002:**
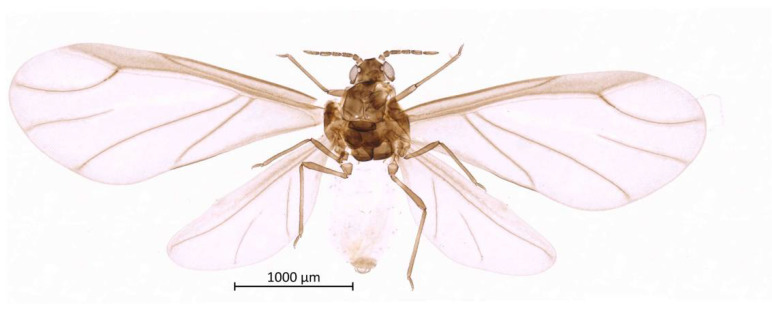
Habitus of alate vivipara holotype of *Qiao jinshaensis* sp. nov. (IOZ(E)54809).

**Figure 3 insects-13-01104-f003:**
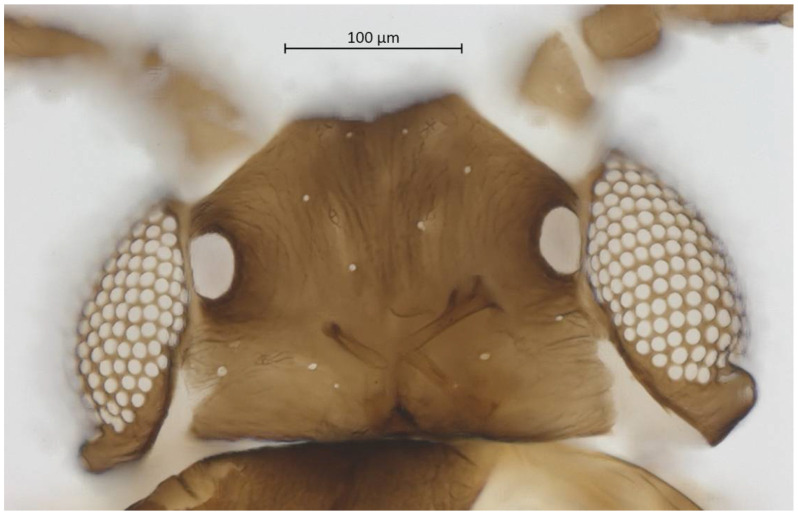
Dorsal head of alate vivipara holotype of *Qiao jinshaensis* sp. nov. (IOZ(E)54809).

**Figure 4 insects-13-01104-f004:**
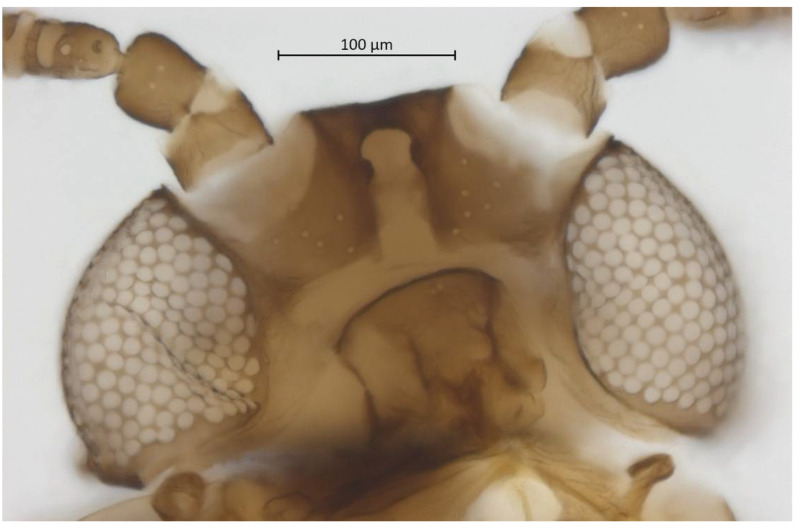
Ventral head of alate vivipara holotype of *Qiao jinshaensis* sp. nov. (IOZ(E)54809).

**Figure 5 insects-13-01104-f005:**
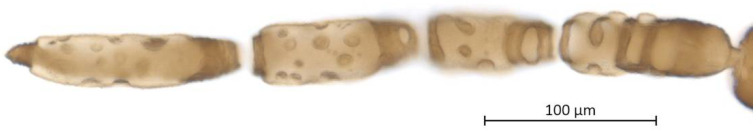
Antenna (ANT III-VI) and sensoria of *Qiao jinshaensis* sp. nov. (QMOR66201).

**Figure 6 insects-13-01104-f006:**
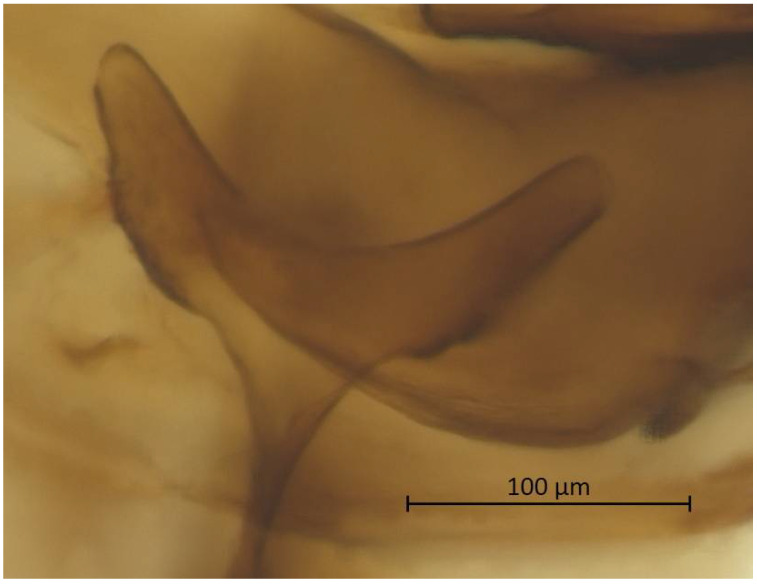
Mesosternal furca of alate vivipara of *Qiao jinshaensis* sp. nov. (QMOR66202).

**Figure 7 insects-13-01104-f007:**
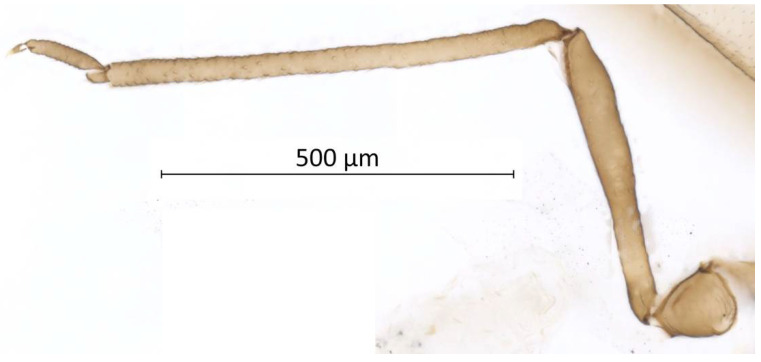
Hind leg of alate vivipara of *Qiao jinshaensis* sp. nov. (QMOR66209).

**Figure 8 insects-13-01104-f008:**
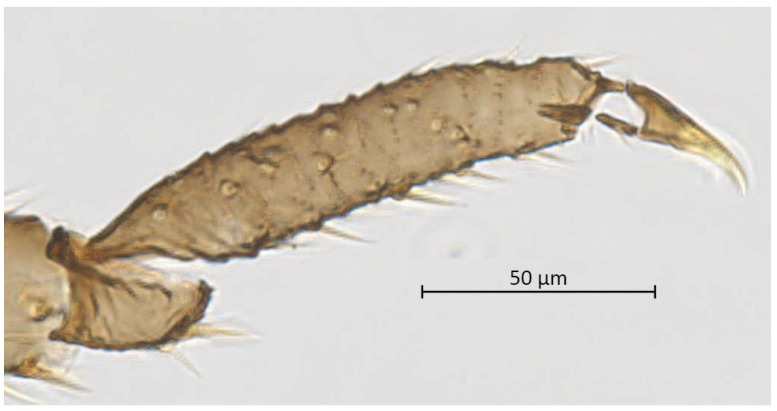
Hind tarsus of alate vivipara of *Qiao jinshaensis* sp. nov. (QMOR66209).

**Figure 9 insects-13-01104-f009:**
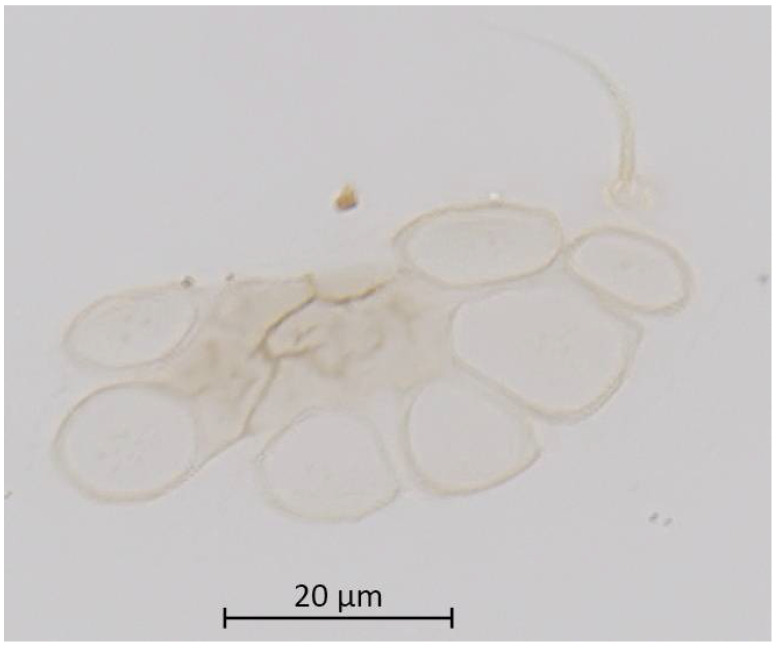
Dorsal abdominal wax gland and single seta of alate vivipara of *Qiao jinshaensis* sp. nov. (IOZ(E)54810).

**Figure 10 insects-13-01104-f010:**
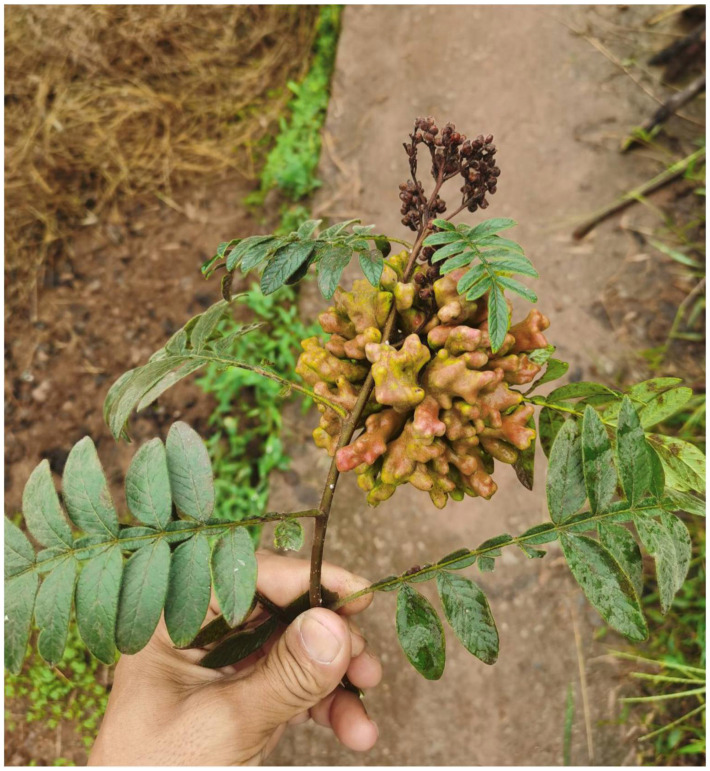
Galls formed by *Qiao jinshaensis* sp. nov. on *Rhus wilsonii*.

**Table 1 insects-13-01104-t001:** Species compared in this study with their GenBank accession numbers for COI and EF1α genes.

	GenBank Accession Numbers		Percent Identity with New Species	
Species or Subspecies	COI	EF1α	COI	EF1α
* Floraphis choui * Xiang	MF043980	MF152697	88.9	93.4
*Floraphis meitanensis* Tsai & Tang	MF043990	MF152698	88.9	93.4
*Kaburagia rhusicola ensigallis* (Tsai & Tang)	MF043984	MF152699	90.2	93.8
* Kaburagia rhusicola ovatirhusicola * Xiang	MF043985	EU363676	90.2	93.3
*Kaburagia rhusicola ovogallis* (Tsai & Tang)	MF043986	MF159561	90.4	94.3
* Kaburagia rhusicola rhusicola * (Takagi)	MF043987	MK424021	90.3	93.6
* Meitanaphis elongallis * Tsai & Tang	MF043989	MF152700	87.6	94.0
* Meitanaphis flavogalis * Tang	MF043982	MK424022	90.6	94.3
* Meitanaphis microgallis * Xiang	MK948431	EU363674	87.9	93.0
* Melaphis rhois * (Fitch)	KY624581	MF159562	90.3	94.9
* Nurudea ibofushi * Matsumura	MF043981	MK424020	88.9	93.9
* Nurudea shiraii * (Matsumura)	MF043978	MF152701	88.5	94.9
* Nurudea yanoniella * (Matsumura)	MF043983	MK424024	88.5	94.7
* Schlechtendalia chinensis * (Bell)	KX852297	EU363670	89.0	93.7
* Schlechtendalia peitan * (Tsai & Tang)	MF043979	MF159563	89.5	94.2

**Table 2 insects-13-01104-t002:** Simple and compound diagnostic characters of *Qiao jinshaensis* sp. nov. for COI locus when compared with other *Rhus*-gall aphids.

Position	10–11	13	15	21–22	24	69	201	331–332	345	405	471	585	606	612	616	712–714	719
* Qiao jinshaensis * sp. nov	TG	G	A	AC	A	T	A	AT	C	C	T	T	C	T	C	TGG	G
Other * Rhus * -gall aphids	AT	T	T	TA	T	A	T/C	TC	T	T	A	A/C	T/A	A	T	CCTCCA	T

**Table 3 insects-13-01104-t003:** Measurements (in μm) of alate viviparae (n = 14) of *Qiao jinshaensis* sp. nov.

Length	Mean (μm)	Range (μm)
BL	1879	1676–2197
Head setae	17	15–20
ANT	531	499–565
ANT I	60	50–64
ANT II	46	42–49
ANT III	99	91–108
ANT III BW	11	9–12
ANT IV	83	75–89
ANT IV sensoria	45	30–53
ANT V	101	95–115
ANT V sensoria	74	65–90
BASE	127	112–139
PT	16	11–20
ANT VI sensoria	94	76–106
URS	69	63–75
LMF	417	385–434
LMT	603	569–657
WMT	32	28–35
HT I	32	29–38
LMTS	17	15–19
HT II	111	103–119
Cauda	36	20–48
Cauda BW	78	57–91

## Data Availability

DNA sequence data are available at GenBank; specimen collection data are available from Hébert et al. [22]; specimens are available at the Institute of Zoology of the Chinese Academy of Sciences and the Ouellet-Robert Entomological Collection of the University of Montreal.

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
