# Peer review of "A New Genus and Species of Gall-Forming Fordini (Hemiptera: Aphididae) on Rhus wilsonii Hemsl. from Yunnan, China"

_insects, 2022, doi:10.3390/insects13121104_

Round 1

Reviewer 1 Report

The article presents important results that deserve to be published in Insects. However, before final acceptance, I suggest the modifications below:

Line 13

Change “gen. nov. et sp. nov.” for “gen. et sp. nov.”

Line 18

Include "Qiao gen. nov." and "Qiao jinshaensis sp. nov.” in keywords.

Line 26

Change “Nurudea ibofushi Matsumura” for “Nurudea ibofushi Matsumura, 1917”

Change “Meitanaphis flavogallis Tang” for “Meitanaphis flavogallis Tang, 1978”

Line 27

Change “Schlechtendalia” for “Schlechtendalia Lessing, 1830”

Change “Kaburagia” for “Kaburagia Takagi, 1937”

Line 33

Change “gen. nov. et sp. nov.” for “gen. et sp. nov.”

Line 39

Include pickup location details (including geographic coordinates)

Line 50

Change “PCR” for “polymerase chain reaction (PCR)”

Lines 56 and 57

Accession numbers OP536221 and OP548621 are not available from Genbank. Authors should review the deposit of sequences.

Line 63

Change “Acyrthosiphon pisum (Harris)” for “Acyrthosiphon pisum (Harris & M., 1776)”

Lines 69 and 70

“…as well as the original published descriptions of other Rhus-gall aphids, were used to assess the identity of the new species.”

The manuscripts must be cited.

Line 101

Change “gen. nov. et sp. nov.” for “gen. et sp. nov.”

Lines 102 and 103

"The genus and species are established together, in accordance with Article 13.4 of 102 the International Code of Zoological Nomenclature (4th Edition).”

Cite the ICZN in the references.

Line 105

Include the sex of the holotype and paratypes.

Line 119

I suggest that authors develop a dichotomous key for genres.

Lines 128-130

In addition to this information, authors need to include a Bayesian tree for the COI.

Line 209

Authors must, at a minimum, include a conclusion. However, I feel that the authors could include a brief discussion of the taxonomy of aphids (for example, which species were described based on classical taxonomy, which were described based on integrative taxonomy and the implications of using each one...).

Line 226

None of the references follow the rules of the journal. Authors must review all.

Author Response

Thank you for the helpful comments. We adopted all of them except for the ones that we explain here.

Line 13

Change “gen. nov. et sp. nov.” for “gen. et sp. nov.”

Done.

Line 18

Include "Qiao gen. nov." and "Qiao jinshaensis sp. nov.” in keywords.

Done.

Line 26

Change “Nurudea ibofushi Matsumura” for “Nurudea ibofushi Matsumura, 1917”

Change “Meitanaphis flavogallis Tang” for “Meitanaphis flavogallis Tang, 1978”

Done.

Line 27

Change “Schlechtendalia” for “Schlechtendalia Lessing, 1830”

Change “Kaburagia” for “Kaburagia Takagi, 1937”

Done.

Line 33

Change “gen. nov. et sp. nov.” for “gen. et sp. nov.”

Done.

Line 39

Include pickup location details (including geographic coordinates)

Done.

Line 50

Change “PCR” for “polymerase chain reaction (PCR)”

Done.

Lines 56 and 57

Accession numbers OP536221 and OP548621 are not available from Genbank. Authors should review the deposit of sequences.

The sequences have already been deposited in GenBank and will be available as soon as the manuscript is published.

Line 63

Change “Acyrthosiphon pisum (Harris)” for “Acyrthosiphon pisum (Harris & M., 1776)”

Date added: Acyrthosiphon pisum (Harris, 1776)

Lines 69 and 70

“…as well as the original published descriptions of other Rhus-gall aphids, were used to assess the identity of the new species.”

The manuscripts must be cited.

Done.

Line 101

Change “gen. nov. et sp. nov.” for “gen. et sp. nov.”

Done.

Lines 102 and 103

"The genus and species are established together, in accordance with Article 13.4 of 102 the International Code of Zoological Nomenclature (4th Edition).”

Cite the ICZN in the references.

Done.

Line 105

Include the sex of the holotype and paratypes.

Done.

Line 119

I suggest that authors develop a dichotomous key for genres.

Several genera are in need of revision, as mentioned in the introduction, and current keys to genera do not separate them all cleanly (e.g., Melaphis). We therefore opted to create a species-level couplet that can be inserted into the key to aphid species on Rhus spp. that is currently available at aphidsonworldsplants.info (section 3.2.6). We hope that you agree that this is a good compromise until such a time as someone can properly review the several genera together.

Lines 128-130

In addition to this information, authors need to include a Bayesian tree for the COI.

Because the new species is so distinct, genetically, from the other Rhus-feeding aphid species, the CO1 and EF1α dendrograms were uninformative. The nodes that connected the new species to the others were relatively deep and poorly supported. As such, we are not comfortable making any kind of phylogenetic inference at this time. Of course, the sequences are all available at GenBank, so others can to use them in their phylogenetic analyses.

Line 209

Authors must, at a minimum, include a conclusion. However, I feel that the authors could include a brief discussion of the taxonomy of aphids (for example, which species were described based on classical taxonomy, which were described based on integrative taxonomy and the implications of using each one...).

We inserted a short conclusion (section 4) that discusses some of the evolutionary and diversification scenarios of the Rhus-feeding and Pistacia-feeding aphids.

Line 226

None of the references follow the rules of the journal. Authors must review all.

Done.

Reviewer 2 Report

Dear Authors,

I have no fundamental comments on your manuscript, but I would give one recommendation for improving the manuscript. Is it possible to give in your manuscript a key to species identification, taking into account the described taxon? That would be useful. As a possible option, a table for comparing morphological characters of a new taxon and those close to it. Is this currently possible?

Lines 110 and 192. I assume that there is not necessary to highlight all the letters in the authors' names in capital letters.

Author Response

Thank you for your helpful comments.

I have no fundamental comments on your manuscript, but I would give one recommendation for improving the manuscript. Is it possible to give in your manuscript a key to species identification, taking into account the described taxon? That would be useful. As a possible option, a table for comparing morphological characters of a new taxon and those close to it. Is this currently possible?

The idea of including a key is a good one. However, rather than providing a comprehensive key to species, we've opted to insert a new couplet into the existing key, available at aphidsonworldsplants.info, which is the key almost everyone uses. Our suggested couplet will no doubt be inserted into the online version of AWP in due time.

Lines 110 and 192. I assume that there is not necessary to highlight all the letters in the authors' names in capital letters.

Regarding capitalized surnames in the manuscript text, we'd like to keep clear which name is a family name versus which is a given name, especially because the order can be different depending on context, as is notably the case with Chinese names.